# Preclinical Long-Term Stability and Forced Degradation Assessment of EPICERTIN, a Mucosal Healing Biotherapeutic for Inflammatory Bowel Disease

**DOI:** 10.3390/pharmaceutics17020259

**Published:** 2025-02-15

**Authors:** Wendy M. Kittle, Micaela A. Reeves, Ashley E. Fulkerson, Krystal T. Hamorsky, David A. Morris, Kathleen T. Kitterman, Michael L. Merchant, Nobuyuki Matoba

**Affiliations:** 1Department of Pharmacology and Toxicology, School of Medicine, University of Louisville, Louisville, KY 40202, USA; wendy.cecil@louisville.edu (W.M.K.); reevesm@hanover.edu (M.A.R.); michael.merchant@louisville.edu (M.L.M.); 2Brown Cancer Center, School of Medicine, University of Louisville, Louisville, KY 40202, USA; 3Center for Predictive Medicine, School of Medicine, University of Louisville, Louisville, KY 40202, USA; dmorris@forgebiologics.com (D.A.M.); kathleen.kitterman@louisville.edu (K.T.K.); 4Department of Medicine, School of Medicine, University of Louisville, Louisville, KY 40202, USA; 5Core and Clinical Proteomics Laboratories, University of Louisville, Louisville, KY 40202, USA

**Keywords:** cholera toxin B subunit, EPICERTIN, long-term stability, forced degradation, drug development, biopharmaceuticals, inflammatory bowel disease

## Abstract

**Background/Objectives:** EPICERTIN, a biotherapeutic candidate for mucosal healing in inflammatory bowel disease (IBD) and other mucosal disorders, was subjected to an extensive long-term stability program to evaluate its molecular stability and physicochemical properties. Additionally, a forced degradation assessment was conducted to identify EPICERTIN’s degradation products under various conditions, including thermal stress, pH variations, agitation, and oxidation. **Methods:** The stability of EPICERTIN drug substance (DS), formulated in phosphate-buffered saline (PBS) at 1 mg/mL and stored at 5 °C and 25 °C/60% relative humidity (RH), was monitored over a 2-year period, referencing relevant regulatory guidelines. Evaluations of EPICERTIN DS over the 24-month period included assessment of purity by SDS-PAGE and size exclusion high performance liquid chromatography (SEC-HPLC), identity by electrospray ionization mass spectrometry (ESI-MS) intact mass analysis and Western blotting, and potency by GM1-binding KDEL-detection ELISA (GM1/KDEL ELISA). The forced degradation patterns were analyzed by assessing purity (using SEC-HPLC and SDS-PAGE), potency (via GM1/KDEL ELISA), and intact mass (via ESI-MS). **Results:** The results overall support that EPICERTIN DS remains stable for 2 years under the tested conditions. The forced degradation assessment effectively identified degradation products, particularly under conditions of high temperatures (above 40 °C for 24 h), low pH values (pH 1 and 4), and oxidation upon exposure to 2% H_2_O_2_. **Conclusions:** These findings highlight EPICERTIN’s robust long-term stability in PBS formulation, reinforcing its potential as a viable drug candidate for the treatment of IBD.

## 1. Introduction

Inflammatory bowel disease (IBD) represents a group of disorders characterized by chronic and relapsing inflammation in the gastrointestinal (GI) tract. The two primary forms of IBD are Crohn’s disease (CD) and ulcerative colitis (UC). CD manifests as non-sequential transmural inflammation anywhere in the GI tract, while UC predominantly affects the inner layers of the distal GI tract in a continuous manner, typically beginning at the rectum and extending into the colon [1]. The treatment regimens for UC and CD are similar in employing a step-up therapeutic approach that escalates with disease severity or when agents lose or lack efficacy [2,3,4]. Current treatments for IBD, including aminosalicylates, corticosteroids, immunomodulators, and biologics, are often associated with limitations such as adverse effects, inconsistent efficacy, and the development of drug resistance [2,3,4]. These shortcomings highlight the urgent need for novel therapeutic approaches that can provide sustained disease control and prevent the progression to severe complications. Given the absence of a curative medication for IBD, mucosal healing represents the most critical clinical endpoint, as it is associated with sustained clinical remission, improved quality of life, and reduced risk of epithelial neoplasia [5,6,7]. Effective interventions that promote mucosal healing therefore have the potential to substantially reduce the need for surgical procedures and mitigate the risk of long-term complications, including IBD-associated colorectal cancer [8,9]. Mucosal healing refers to the restoration of the colonic mucosa’s epithelial barrier to a healthy, homeostatic state. Epithelial repair is a critical step in mucosal healing, as recovering the intestinal epithelial barrier inhibits inflammation caused by the entry of bacteria into the mucosa. Thus, an agent that can restore the epithelial barrier and facilitate mucosal healing without suppressing immune function may fill a current treatment gap for IBD. However, there are currently no targeted mucosal healing agents available on the market.

We previously reported that oral administration of EPICERTIN (epithelial cell ER-targeted protein), a recombinant variant of cholera toxin B subunit (CTB), facilitates epithelial repair and mucosal healing in dextran sulfate sodium (DSS)-induced acute and chronic colitis mouse models [10,11]. CTB is the nontoxic homopentameric component of the cholera toxin with high binding affinity to GM1 ganglioside on epithelial cells [12]. EPICERTIN was modified with a C-terminal hexapeptide extension containing a KDEL endoplasmic reticulum (ER) retention motif [13,14]. While both CTB and EPICERTIN bind to GM1 for cellular entry, only EPICERTIN induces mucosal healing in the DSS colitis model [15]. EPICERTIN’s unique mucosal healing activity is attributed to the molecule’s capacity to interact with the KDEL receptor and subsequently activate the inositol-requiring enzyme 1/X-box binding protein 1 arm of an unfolded protein response (UPR) in colon epithelial cells. The UPR is a conserved cellular response to ER stress, promoting cellular adaptation and survival [16] as well as providing novel therapeutic potential for IBD [17]. This UPR-mediated mechanism of action distinguishes EPICERTIN from immunosuppressive strategies typically employed in IBD treatment citation. Additionally, EPICERTIN can be administered topically to the colon or by oral gavage to alleviate DSS-induced colitis in mice [15]. While oral administration of EPICERTIN requires prerequisite neutralization of stomach acid, we previously addressed this limitation by developing a prototype enteric-coated oral formulation for pH-dependent release of EPICERTIN in the colon [18]. Further evidence for EPICERTIN’s effectiveness came from studies on human colon tissues obtained from IBD patients undergoing total colectomy in which ex vivo assays showed EPICERTIN induced wound healing responses as indicated by the presence of viable, proliferating Ki67 positive crypts, whereas these effects were not observed with CTB or PBS treatments [15]. Notably, EPICERTIN treatment significantly increased the expression of multiple genes associated with wound healing (including *TGFB1*, *CDH1*, and *WNT5A*) in colectomy tissues from various IBD patients, for which the most pronounced effect was observed in tissue from a 20-year-old male UC patient, where EPICERTIN upregulated the expression of 79 out of 84 wound healing-related genes [15].

EPICERTIN’s current developmental stage involves the preclinical generation of appropriate chemistry, manufacturing, and controls (CMC), following regulatory guidance from the Food and Drug Administration (FDA) and referencing the International Council for Harmonisation (ICH). CMC encompasses the scientific understanding, large-scale production, and quality control procedures that ensure the safety and efficacy of a drug product. This involves defining the drug’s structure, developing a controlled manufacturing process, and implementing robust analytical methods to consistently produce a high-quality drug substance and drug product. Stability testing is a crucial task at this stage of development. According to ICH Q1A(R2) guidance, the purpose of stability testing a drug is twofold: to demonstrate how the quality of the drug is affected upon exposure to a variety of environmental factors and to establish a retest period with recommended storage and shelf-life conditions [19]. Stability of a drug is assessed by a comprehensive stability testing program evaluating its physicochemical, functional, and basic attributes such as identity, purity, potency, and safety, all of which inform specifications that the drug must meet for future stages of development [20].

Stability testing is not performed using a “one size fits all” approach; the intended shelf life and storage conditions of a specific drug determine what stability conditions must be tested. If long-term studies are conducted, a minimum of 12 months of stability data, with timepoints at three-month intervals, are required at the time of investigational new drug (IND) application submission [19]. The storage conditions of the drug should be tested for stability at 25 °C and 60% relative humidity (RH). If the drug is intended for storage in a refrigerator, stability at 5 °C should also be assessed [19]. If a drug is found to be exceptionally stable at 12 months, stress testing may be utilized to identify likely degradation pathways and substantiate the stability-indicating power of the analytical procedures used [19]. By unveiling a drug’s susceptibility to potential stressors, forced degradation studies significantly enhance the understanding of a drug’s molecular properties [21,22] and provide crucial information necessary for securing regulatory approval.

Stability studies, as outlined in ICH guidelines (ICH Q1A(R2) and Q5C), are crucial for evaluating drug substance quality under various conditions. Particularly, forced degradation studies elucidate degradation pathways under stress conditions (e.g., acidic, basic, oxidative, thermal, etc.) and demonstrate the suitability of analytical methods for quality control. This information is crucial for developing robust analytical methods that can accurately quantify the drug substance even in the presence of degradation products. This ensures patient safety by enabling the accurate quantification of drug substances. This rigorous testing exposes potential degradation pathways impacting purity and potency, guiding formulation and packaging optimization, and predicting long-term stability under normal conditions [21,22]. Ultimately, this helps ensure patient safety while informing the development of reliable analytical methods [21,22]. To this end, we evaluated the preclinical stability profile of research-grade EPICERTIN drug substance (DS) in PBS at 1 mg/mL at 5 °C and 25 °C/60% RH over a long-term 2-year period as well as under forced degradation conditions, which yielded key CMC information to facilitate a future formal stability assessment and first-in-human (FIH) clinical trial.

## 2. Materials and Methods

Production of EPICERTIN. EPICERTIN drug substance (DS) was produced in *Nicotiana benthamiana* plants and purified to >95% homogeneity with an endotoxin level of <1 endotoxin units per mg, as described previously [23], but with the addition of an anion chromatography exchange step using Capto-Q resin prior to metal affinity. EPICERTIN DS was packaged into 5 mL PETG vials with a 2.5 mL fill and a concentration of 1.06 mg/mL in PBS. Purity (SDS-PAGE and SEC-HPLC), identity (ESI-MS), and potency (GM1/KDEL ELISA) were verified. EPICERTIN material used for analytical reference standards was freshly prepared following the same production and purification regimen.

EPICERTIN Biochemical Characterization Toolbox. The “toolbox” for EPICERTIN stability assessments consists of test methods to evaluate test parameters such as purity (SDS-PAGE and size exclusion high performance liquid chromatography (SEC-HPLC)), identity (electro-spray ionization mass spectrometry (ESI-MS) and Western blot (WB)), and potency (GM1-capture KDEL-detection (GM1/KDEL) ELISA). To assess stability, specifications and acceptance criteria for each test method were established and justified based on multiple EPICERTIN batch measurements at release as advised by ICH guidelines (Q6A) and Pre-IND communication with the FDA. The detailed methods for EPICERTIN’s test methods and specifications are described below and can also be referenced in brevity in Table 1.

SDS-PAGE. Reducing SDS-PAGE was performed to evaluate purity, using a pre-cast Mini-PROTEAN TGX 4–20% gradient gel (Bio-Rad, Hercules, CA, USA). EPICERTIN samples were diluted into 2X sample buffer containing 4% SDS and 10% BME and incubated at 95 °C for 10 min. Samples were loaded (10 µg), and the gel was run at 200 V for 35 min, Coomassie (R-250) stained for 1 h, then destained, rocking in destain buffer (12% (*v*/*v*) ethanol, 12% (*v*/*v*) acetic acid) for 5 h. Colorimetric images were captured by Amersham Imager 600. Evaluation of monomeric content by densitometry assessment was performed using GelAnalyzer software (Version 19.1), with a specification of ≥95% monomer. Pre-IND communication with the FDA concurred a purity threshold for pentameric EPICERTIN as ≥95% pure by SEC-HPLC assessment. As such, we held the same standard for the purity threshold for EPICERTIN as ≥95% pure by SDS-PAGE assessment.

SEC-HPLC. The percent of intact pentamer EPICERTIN (61.4 kDa) was assessed as described previously [18] to evaluate purity. Briefly, samples at a concentration of 1 mg/mL were run over a Tosoh TSKgel SuperSW300 column using PBS running buffer. The protein molecular weight standard (MSTD) (Calbiochem 539053, Temecula, CA, USA) used contains five highly purified proteins ranging in molecular weight from 12.4 kDa to 290 kDa: cytochrome c, 12.4 kDa; myokinase, 32 kDa; enolase, 67 kDa; lactate dehydrogenase, 142 kDa; and glutamate dehydrogenase, 290 kDa. Evaluation of chromatograms considers retention time in minutes and area under the curve (AUC, in milli-absorbance units (mAu)), with specifications of retention time (16.4 ± 0.2 min) and ≥95% AUC. The AUC specification was informed by pre-IND communication with the FDA, and retention time was redefined based on multiple batch measurement validation after column calibration.

ESI-MS. Samples obtained from freshly opened vials were sent to The Scripps Research Institute (San Diego, CA, USA) for analysis. This intact mass analysis is an EPICERTIN identity method, used to identify the presence and content of C-terminally intact and truncated species as described previously [23]. Intact monomeric mass is assessed (12,280 Da), with a specification of 12,280 ± 3 Da. This specification indicates the theoretical intact mass of an EPICERTIN monomer (12280 Da) as well as a 3 Da range for measurement tolerance that accounts for this acceptable deviation in resolving power of the mass analyzer.

WB. This test method evaluates identity and was implemented in this work a single time to demonstrate the identity of EPICERTIN dimer. SDS-PAGE gels were run according to the procedure described above, with 500 ng protein per sample loaded. Transblot Turbo (BioRad, Hercules, CA, USA) was used for semi-dry transfer to the PVDF membrane (BioRad, Hercules, CA, USA). The membrane was blocked using PBST (1× PBS, 0.05% (*w*/*v*) Tween 20) containing 5% (*w*/*v*) non-fat dry milk (PBSTM), rocking at 4 °C overnight. The membrane was washed 3 times with PBST, 5 min each, rocking. The membrane was stained with a primary antibody (9F9C7 rat anti-CTB mAb [24]), diluted 1:20,000 in 1% PBSTM, for 1 h at room temperature with gentle rocking. The membrane was washed again and then stained with secondary antibody (goat anti-rat IgG-HRP, 3030-05; Southern Biotech, Birmingham, AL, USA) diluted 1:2000 in 1% PBSTM for 1 h at room temperature with gentle rocking. The membrane was washed again, and chemiluminescent detection reagent (Amersham ECL Prime, Cytiva, Wilmington, DE, USA) was added to the membrane according to the manufacturer’s instructions. Chemiluminescent images were captured by Amersham Imager 600 (GE Healthcare Bio-Sciences, Uppsala, Sweden).

GM1/KDEL ELISA. Potency was assessed by a GM1-binding KDEL-detection ELISA as previously described [18,23]. Briefly, plates were coated with 2 μg/mL GM1 ganglioside (Sigma-Aldrich, St. Louis, MO, USA) for 2 h at room temperature (20–23 °C). The plate was washed three times with PBST and blocked with 5% PBSTM overnight at 4 °C. Samples were added at a top concentration of 2 μg/mL in 1% PBSTM, diluted 1:2 across the plate, and then incubated for 2 h at room temperature. The primary antibody (10C3 mouse anti-KDEL monoclonal antibody, 1:1000; Enzo, Farmindale, NY, USA) was applied and incubated at room temperature for 1 h. The secondary antibody (goat anti-mouse IgG-HRP, 1030-05, diluted 1:5000; SouthernBiotech, Birmingham, AL, USA) was then applied and incubated at room temperature for 1 h. TMB (BioFX™ TMB One Component HRP Substrate, Surmodics IVD, Eden Prairie, MN, USA) was used to detect GM1-bound EPICERTIN with the intact KDEL motif. Absorbance was measured at 450 nm in a Synergy HT plate reader (BioTek Instruments Inc., Winooski, VT, USA) after the reaction was stopped. Evaluation of GM1/KDEL ELISA considers a specification of %EC_50_ shift < ±30% relative to the reference standard. This acceptance criterion is generally considered acceptable for developmental phases prior to full validation for use in late-stage clinical studies.

Long-term stability experimental design. EPICERTIN DS was produced as described above and stored for 2 years in a monitored refrigerator or an environment-controlled chamber (CARON model CRSY102 environmental chamber) set at either 5 °C or 25 °C/60% RH. At each time point (0, 1, 3, 6, 9, 12, and 24 months), a new vial from both temperatures was pulled, protein concentration was measured by Thermo Scientific™ NanoDrop™ OneC (Madison, WI, USA), and the pH of each vial was measured using a VWR SB90M5 Benchtop sympHony Meter. Analytical assessments for identity, purity, and potency were performed as follows.

Long-term stability test methods. At each time point, purity (SDS-PAGE; SEC-HPLC), identity (ESI-MS; WB), and potency (GM1/KDEL ELISA) were assessed as described above. Appearance, concentration, physicochemical properties, and safety (contracted microbial enumeration test (USP<61/62>)) were also assessed at each time point. Appearance was evaluated by an acceptance criterion of clear, colorless liquid, free of visible particles (CCLFVP), and concentration was evaluated by an acceptance criterion of 1 ± 0.2 mg/mL. Physicochemical properties and safety were evaluated by acceptance criteria of pH 7.4 ± 0.2 and <10 colony-forming units (CFU)/mL, respectively. These specifications were communicated in the Pre-IND.

Forced degradation stability experimental design. EPICERTIN DS was produced as described above and stored at 5 °C. EPICERTIN DS, formulated in PBS at 1.06 mg/mL, was subjected to several forced degradation conditions: high temperature (40, 50, and 60 °C for 24, 48, and 72 h), freeze–thaw (1, 2, 3, 4, and 5 cycles), pH (pH 1, 4, 7, and 10 for 24 h), agitation (24 and 48 h), and oxidation (0.2% and 2% H_2_O_2_ (*w*/*v*) for 36 h). High temperature: 20 µL of EPICERTIN DS was aliquoted into 2 mL screw cap vials. Vials were placed in water baths set for 40, 50, and 60 °C for 24, 48, and 72 h. Freeze–thaw: 100 µL of EPICERTIN DS was aliquoted into 2 mL screw cap vials and exposed to up to five cycles of −20 °C/25 °C. pH: EPICERTIN DS samples were pH-adjusted using either 1 N NaOH or 1 M HCl to reach final pH values of 1, 4, 7, and 10, and 20 µL of each sample was aliquoted into 1.7 mL microcentrifuge tubes and stored at 5 °C for 24 h. Agitation: 100 µL EPICERTIN DS samples were aliquoted into 1.7 mL microcentrifuge tubes and agitated at room temperature by vortexing at maximum speed (Vortexer 59A, Denville Scientific Inc., South Plainfield, NJ, USA) for 24 and 48 h. Oxidation: EPICERTIN DS was exposed to a final concentration of 0.2% and 2% hydrogen peroxide (H_2_O_2_), and 60 µL aliquots were stored in 1.7 microcentrifuge tubes at 25 °C for 36 h. Analytical assessments of purity, potency, and identity were performed as follows.

Forced degradation test methods. For all forced degradation conditions, purity (SEC-HPLC) and potency (GM1/KDEL ELISA) were assessed as described above. In response to SEC-HPLC results, purity was also assessed by SDS-PAGE for all conditions. Oxidation samples were further assessed by ESI-MS intact mass analysis to evaluate oxidation status as well as by peptide mapping to determine oxidation sites as detailed in the following paragraph.

Peptide mapping. For the determination of oxidation sites, we employed peptide mapping via S-Trap micro spin column-based Glu-C digestion of protein samples: EPICERTIN DS exposed to 2% H_2_O_2_ and an unoxidized reference standard. Samples were diluted in a buffer containing 5% (*w*/*v*) sodium dodecyl sulfate (SDS) (Sigma-Aldrich, St. Louis, MO, USA) and 50 mM triethylammonium bicarbonate (TEA-BC) (Sigma-Aldrich, St. Louis, MO, USA), followed by reduction with 20 mM dithiothreitol (DTT) (Sigma-Aldrich, St. Louis, MO, USA) and alkylation with 40 mM iodoacetamide (Sigma-Aldrich, St. Louis, MO, USA). After acidification with 2.5% (*w*/*w*) phosphoric acid (Sigma-Aldrich, St. Louis, MO, USA) samples were loaded onto S-Trap columns (Protifi, Fairport, NY, USA) and washed with a binding buffer containing 100 mM TEA-BC in 90% (*v*/*v*) methanol and 10% (*v*/*v*) water at pH 7.55. Glu-C digestion was performed overnight at 37 °C. Peptides were eluted sequentially with 50 mM TEA-BC buffer, 0.2% (*w*/*w*) formic acid (Fisher Scientific #A117-50, Waltham, MA, USA), 50% (*v*/*v*) acetonitrile (Fisher Scientific #A955-4, Waltham, MA, USA), and dried using a SpeedVac Vacuum Concentrator (Thermo Scientific, Waltham, MA, USA). The dried peptides were dissolved in a solution containing 2% acetonitrile and 0.1% formic acid, and their concentration was estimated by nanodrop measurement. Samples and technical replicates were acquired by nanoflow liquid chromatography mass spectrometry (nLC–MS). Thermo Proteome Discoverer version 2.5.0.400 was used to process raw nLC-MS data and identify peptide sequences and post-translational modifications.

## 3. Results

### 3.1. Long-Term 2-Year Stability Assessment Results

The 2-year long-term stability assessment is summarized in Table 1, which includes an overview of the implemented test parameters, methods, acceptance criteria, and results of the initial and final timepoints. The complete data set, including all intermediate timepoints, is available in Appendix A for the 5 °C condition and in Appendix A for the 25 °C/60% RH condition.

**Table 1 pharmaceutics-17-00259-t001:** EPICERTIN DS 2-year long-term stability assessment overview: test parameters, methods, acceptance criteria, and initial and final timepoint results.

			0 Months		24 Months			
Test Parameter	Test Method	Acceptance Criteria	Initial		5 °C		25 °C/60% RH	
Appearance	Visible appearance	Clear, colorless liquid, free of visible particles (CCLFVP)	Pass	CCLFVP	Pass	CCLFVP	Pass	CCLFVP
Protein Concentration	A_280_	1 ± 0.2 mg/mL	Pass	1.06 ± 0.004	Pass	1.13 ± 0.025	Fail	1.35 ± 0.032
Physicochemical Properties	pH	7.4 ± 0.2	Pass	7.25	Pass	7.32	Pass	7.25
Safety (Microbial Enumeration)	USP <61> and USP <62>	<10 CFU/mL	Pass	<10 CFU/mL	Pass	<10 CFU/mL	Pass	<10 CFU/mL
Identity	ESI-MS	12,280 ± 3 Da	Pass	12,280	Pass	12,280	Pass	12,281
Purity	Reducing SDS-PAGE	≥95% EPICERTIN	Pass	100.00%	Pass	100.00%	Pass	100.00%
Purity	SEC-HPLC	16.4 ± 0.2 min retention time and ≥95% AUC	Pass	100.00%	Pass	100.00%	Pass	100.00%
Potency	GM1/KDEL ELISA	EC50 shift <±30% of reference standard	Pass	+2.85%	Pass	−9.6%	Pass	−26.1%

#### 3.1.1. Identity

ESI-MS intact mass analysis was used to detect degradation of EPICERTIN DS. At the initial timepoint, the intact mass profile showed a major peak at the expected molecular mass for EPICERTIN monomer (~85% 12,280 Da, -KDEL intact) with two minor peaks at the expected molecular masses for C-terminally truncated Leu (~5% 12,038 Da, -KDE intact) and Glu-Leu species (~10% 12,167 Da, -KD intact). As a result of the platform-endogenous carboxypeptidases, these truncated species are consistently present and expected in these proportions in plant-made EPICERTIN DS [13,23].

This method showed that the EPICERTIN DS mass profile remained unchanged at 24 months at both 5 °C and 25 °C/60% RH and maintained proportions of C-terminally truncated species consistent with the initial time point (Figure 1). Additionally, an anti-CTB WB of SDS-PAGE confirms the identity of EPICERTIN DS (Figure 2). 

#### 3.1.2. Purity

The purity of EPICERTIN DS was assessed by SDS-PAGE and SEC-HPLC. Reducing SDS-PAGE (Figure 2) was used to assess the purity of EPICERTIN and demonstrated no alterations over the two-year period for EPICERTIN DS stored at either 5 °C or 25 °C/60% RH. Anti-CTB WB was performed to confirm the identity of EPICERTIN dimer bands around 25 kDa. Purity was also assessed by SEC-HPLC, and the native pentamer conformation of EPICERTIN DS was unaffected upon storage at 5 °C for 24 months (Figure 3C), with a retention time of 16.37 min 100% AUC. Slight degradation of pentamer (16.29 min, 97.5% AUC) was observed after storage at 25 °C/60% RH for 24 months, as indicated by the presence of a small peak at a higher retention time (20.95 min, 2.5% AUC) (Figure 3D).

#### 3.1.3. Potency

The GM1/KDEL ELISA, a surrogate biological potency assay [23], was used to gauge the stability of the EPICERTIN DS molecular functions. This assay quantifies GM1-binding activity and confirms the integrity of the C-terminal ER retention motif, both of which are essential for EPICERTIN’s mucosal healing activity [15]. Stability samples were assessed using their nominal concentration of 1.06 mg/mL for the ELISA. After 24 months of storage, the EC_50_ of EPICERTIN DS remained within the specification at 5 °C and at 25 °C/60% RH (<±30% EC_50_ shift), as shown in Figure 4. However, at 25 °C/60% RH, there was an increase at the 12-month timepoint, with a %EC_50_ shift of −38.8%, likely reflecting the increased concentration of EPICERTIN over time under this storage condition (Appendix A). Overall, the results of the long-term stability assessment suggest that EPICERTIN DS retains its stability for at least up to 24 months under the tested conditions. A comprehensive summary of results from the long-term stability assessment can be found in Appendix A.

### 3.2. Forced Degradation Assessment

The stability parameters used to evaluate EPICERTIN DS degradation products under forced degradation conditions included purity (SEC-HPLC and SDS-PAGE) and potency (GM1/KDEL ELISA) for all conditions previously described, with an additional identity (ESI-MS) assessment for the oxidation conditions. All acceptance criteria were the same as the long-term study, which are summarized in Table 1. Additionally, peptide mapping was performed for oxidation site determination.

#### 3.2.1. Purity

##### SEC-HPLC

Conditions failing to meet specifications by SEC-HPLC included high temperature (40 °C 48 h, 40 °C 72 h, 50 °C 24 h, 50 °C 48 h, 50 °C 72 h, 60 °C 24 h, 60 °C 48 h, and 60 °C 72 h) and low pH (pH 1 and pH 4), with representative chromatograms from each condition shown in Figure 5. All other conditions met specifications. Alongside EPICERTIN pentamer, high temperature conditions generally showed both higher and lower molecular weight species on SEC-HPLC chromatograms, which could correspond to higher-order structures (e.g., decamers and higher-order aggregates) and lower-order structures (e.g., monomer, dimer, etc.). Low pH conditions showed lower molecular weight species on SEC-HPLC chromatograms, which likely correspond to EPICERTIN monomer and dimer [13,18]. The SEC-HPLC chromatograms of high temperature conditions at the 24 h time point are available in Appendix A.

##### SDS-PAGE

To further characterize the degradation patterns, we employed SDS-PAGE for all forced degradation conditions. All conditions met the specification of ≥95% EPICERTIN on SDS-PAGE (Appendix A).

#### 3.2.2. Potency

Conditions that failed to meet the GM1/KDEL ELISA specification included low pH (pH 1 and 4), as shown in Figure 6. All other conditions met the specification. Potency was attenuated with pH 4 (+335.9% EC_50_ shift) and negligible with pH 1 (+34,128.5% EC_50_ shift).

#### 3.2.3. Identity

ESI-MS was employed to detect evidence of H_2_O_2_ oxidation, which occurs in mass-increase intervals of 16 Da [21,25,26,27]. The lower strength oxidation condition, 0.2% H_2_O_2_ for 36 h, met the specification and remained unoxidized. However, the higher strength oxidation condition, 2% H_2_O_2_ for 36 h, was sufficient to oxidize EPICERTIN DS, as evidenced by a 16–17 Da increase in mass compared to an unoxidized reference standard by ESI-MS analysis (Figure 7).

H_2_O_2_ is commonly used to probe for oxidation in forced degradation assessments, and free cystine (Cys, C) and methionine (Met, M) are the most frequently oxidized residues [21,25,26,27]. In EPICERTIN monomer, the two Cys residues form a disulfide bridge, thus leaving one of the three Met residues to be the likely site of the H_2_O_2_ oxidation. To determine which methionine residue was oxidized, we performed peptide mapping (Figure 8). The results revealed that 100% of Met37 and 54% of Met101 were oxidized, while negligible oxidation was observed at Met68 upon 36 h exposure to 2% H_2_O_2_. Approximately 40% of Met37 and 32% of Met101 were also oxidized in the reference standard not exposed to H_2_O_2_, likely reflecting an artifact of sample processing and/or acquisition during the peptide mapping analysis. Though EPICERTIN DS was oxidized, this modification does not affect GM1 binding activity or KDEL intactness, which are essential for EPICERTIN’s mechanism (Appendix A, Appendix A), as the acceptance criterion for GM1/KDEL ELSIA was met. Additionally, oxidation does not seem to affect EPICERTIN’s conformational stability, as the oxidized EPICERTIN DS met the specifications for both SDS-PAGE and SEC-HPLC (Appendix A).

Overall, the results showed that EPICERTIN DS degraded under conditions of high temperature (above 40 °C for 24 h), as indicated by SEC-HPLC (purity), as well as low pH (pH 1 and 4), as demonstrated by SEC-HPLC and GM1/KDEL ELSIA (potency). EPICERTIN DS was oxidized with 2% H_2_O_2_ as determined by ESI-MS. The complete forced degradation assessment is summarized in Appendix A.

## 4. Discussion

The goal of this work is to advance the development of EPICERTIN as a novel mucosal healing biotherapeutic for IBD treatment. The stability assessments performed here address FDA requirements for the generation of CMC data at the preclinical stage, which are necessary to advance EPICERTIN to a formal regulatory-compliant stability study and FIH trial.

The long-term stability results indicate that the EPICERTIN DS, formulated as a 1.06 mg/mL solution in PBS, remains stable upon storage at 5 °C for up to 24 months. During storage at both 5 °C and 25 °C/60% RH, the native pentameric stability remains within specifications in all cases, with a noted slight (<3%) degradation upon storage at ambient temperatures (25 °C/60% RH; Figure 3D). Protein concentration showed a notable deviation from specification for EPICERTIN DS stored at 25 °C/60% RH, with measured concentrations exceeding the specification beginning at the 9-month timepoint (Appendix A). Further investigation revealed that the concentration deviation was due to an inadequate container-closure system. Despite employing an industry-standard packaging option intended to provide an airtight seal, we observed unexpected concentration deviations. Subsequent investigation revealed that the packaging did not maintain a complete seal, leading to evaporation over time and a gradual increase in protein concentration. The observed concentration changes highlight the need for packaging that provides a more robust barrier against evaporation. During future clinical development and large-scale production, we will conduct extensive compatibility studies under various environmental conditions to ensure consistent protein concentration and potency throughout the product’s shelf life. These studies will evaluate a wider range of industry-standard packaging materials, including those incorporating rubber stoppers or septa, as well as optimizing fill volumes and headspace. For the 25 °C/60% RH samples that exceeded the concentration acceptance criterion, the GM1/KDEL ELSIA results showed a discordancy: the 9-month and 24-month samples were in specification, yet the 12-month sample did not meet the specification. Overall, the EPICERTIN DS was found to be stable upon storage at 5 °C. Under 25 °C/60% RH conditions, while minor degradation within specification was observed by SEC-HPLC at the 24-month timepoint, and an evaporation issue caused by the container closure system at later timepoints was noted, EPICERTIN DS maintained its overall stability under the tested conditions.

In the forced degradation study, conditions leading to EPICERTIN DS degradation were identified. Degradation species were detected by SEC-HPLC (Figure 5) for several conditions: high temperature (all conditions above 40 °C for 24 h) and low pH (pH 1 and 4). Degradation species were also detected by GM1/KDEL ELISA (Figure 6), specifically for low pH (pH 1 and 4), and with ESI-MS (Figure 7) for oxidation (2% H_2_O_2_). Regarding EPICERTIN purity, residual dimer bands on SDS-PAGE were confirmed by WB (Figure 2), indicating an incomplete reduction of EPICERTIN pentamer into monomer. Accordingly, the presence of dimer bands on SDS-PAGE gels reflects a technical artifact of the method rather than a quality loss.

Interestingly, despite detecting new degradation patterns by the SEC-HPLC assessment, out of all forced degradation conditions tested, only low pH conditions (pH 1 and 4) affected potency, as indicated by the attenuated GM1 binding observed with the GM1/KDEL ELISA (Figure 6). This finding agrees with previous studies demonstrating that EPICERTIN degrades into monomers upon exposure to low pH [13,18]. We have initiated formulation development to protect EPICERTIN during administration from denaturation by stomach acid, using enteric-coated capsules, in which EPICERTIN in PBS + 100 mM mannitol was spray-dried and encapsulated within a gelatin capsule coated with an anionic polymer for pH-dependent release in the colon [18]. The forced degradation study also identified degradation species from high temperature exposure by SEC-HPLC that were undetected by the GM1/KDEL ELISA (Appendix A). We speculate that the GM1/KDEL ELLISA assay conditions, employing the use of Tween 20 detergent, could help resolve aggregation species, as Tween 20 has been found to enhance CTB folding and assist in the oligomerization of CTB to the pentameric form [29]. Given the limitation of the GM1/KDEL ELISA in detecting the high temperature degradation products identified by SEC-HPLC, as well as a discordant result observed with the 25 °C/60% RH samples, which failed the concentration specification due to evaporation, we acknowledge the need to further optimize and refine the assay for method validation. Additionally, the development of an orthogonal assay to monitor potency may be necessary to more accurately measure potency.

Forced degradation assessments were conducted as part of this foundational investigation using research-grade material to identify potential degradation products as well as to develop and challenge the stability-indicating power of analytical methods, consistent with ICH guidelines. These studies aimed to understand degradation pathways under various stress conditions. We acknowledge that the different sample volumes used in these forced degradation studies resulted in varying surface area-to-volume ratios, which could influence degradation kinetics by altering oxygen exposure, compared to the long-term stability assessment. Future studies using clinical-grade material manufactured under Good Manufacturing Practices (GMP) conditions will focus on quantifying degradation kinetics under relevant storage conditions and in appropriate container closure systems.

Importantly, the preclinical stability assessments presented here will serve as the cornerstone for establishing a robust and regulatory-compliant formal stability assessment framework. This is essential because defining formal specifications (i.e., test parameters, methods, and acceptance criteria) mandates supporting data from preclinical and/or clinical studies to justify their selection, as per ICH Q6B guidelines. Currently, the data presented primarily exhibit a qualitative nature, which is typical at this stage of development. We recognize that the next steps involve a comprehensive validation of analytical methods. This validation will also be instrumental in determining the necessary extent of repeated measures and the appropriate statistical analysis to confidently establish shelf life and release specifications, aligning with ICH Q1A(R2) and Q1E guidance. While the present study demonstrated the overall outstanding stability of EPICERTIN DS in PBS, further optimization of storage conditions may be considered for product development if necessary. For instance, protection from aggregation resulting from high temperature could be necessary for the logistical transport of the drug product. This could be accomplished through the use of excipients such as salts or detergents. The forced degradation study revealed that oxidation can occur under extreme conditions, likely at Met37 and possibly at Met101 (Figure 8). To prevent this, an antioxidant could be added to the EPICERTIN formulation. We could also consider targeted interventions, such as introducing a specific mutation, to protect against the potential for oxidation. For example, Q-griffithsin, a clinically studied broad-spectrum antiviral biotherapeutic, was protected from oxidation by a M78Q mutation following formulation stability assessments that identified this site as prone to oxidation [30]. Nevertheless, ESI-MS analysis clearly demonstrated that EPICERTIN DS is not susceptible to chemical modifications such as deamidation and oxidation under the intended and accelerated storage conditions (5 °C and 25 °C/60% RH). Furthermore, neither potency nor conformational stability of EPICERTIN was affected by the forced oxidation conditions with 2% H_2_O_2_ exposure (Appendix A), suggesting that oxidation may not pose a significant risk to the quality of EPICERTIN DS. In summary, EPICERTIN DS exhibits excellent stability under the tested conditions, and these findings provide a strong foundation for future GMP development and optimization of its formulation and storage.

From a regulatory perspective, guidance authorities are interested in the identification of drug degradation products since such alterations in pharmaceutical quality attributes could affect efficacy and safety profiles [21,22]. EPICERTIN’s “toolbox” provides orthogonal test methods that permit the assessment of DS stability over time and under forced degradation conditions. The forced degradation results presented herein successfully identified conditions that produce degradation products, as determined by our stability-indicating test methods, thereby demonstrating the suitability of their stability-indicating power. These results justify the use of these methods for the long-term stability study as well as for future clinical-grade EPICERTIN DS manufacturing processes. As in the case of EPICERTIN, all drugs on the path to the clinic should have a tailored “toolbox” of test methods capable of detecting degradation products to inform the drug’s stability and quality profiles, all of which are interrogated by regulatory agencies. Further optimization and refinement of assay protocols and specifications are underway to qualify and validate these methods in support of a future formal stability assessment as well as EPICERTIN’s clinical development.

EPICERTIN is a promising first-in-class mucosal healing biotherapeutic with the potential to enhance the UC standard of care. Achieving mucosal healing is an important clinical goal for remission [5,6,7], and EPICERTIN could offer an earlier intervention option or serve as a complementary combination therapy to improve the long-term outcomes for patients. Following the establishment of a successful clinical proof of concept in UC patients, the potential for targeting CD can be further explored as well. The long-term stability and forced degradation assessments of EPICERTIN DS conducted in this study will provide key regulatory information required to prepare an IND submission to the FDA, which is a crucial milestone in EPICERTIN’s path toward a FIH clinical trial for mucosal healing in IBD patients.

## Figures and Tables

**Figure 1 pharmaceutics-17-00259-f001:**
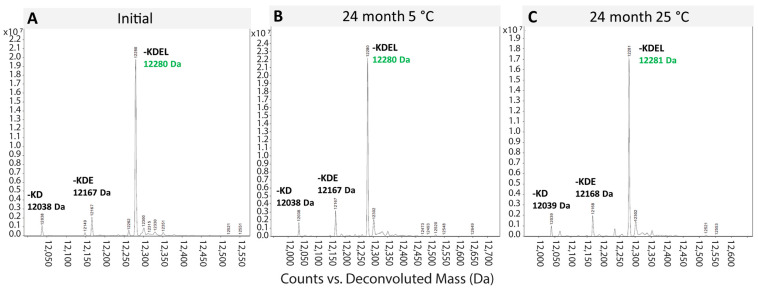
ESI-MS scan of EPICERTIN DS at 24 months. (**A**) ESI-MS chromatogram demonstrating EPICERTIN identity by monomeric intact mass within specification (12,280 ± 3 Da), at the initial time point with unchanged truncated content both (**B**) after 24 months at 5 °C and (**C**) after 24 months at 25 °C/60% RH.

**Figure 2 pharmaceutics-17-00259-f002:**
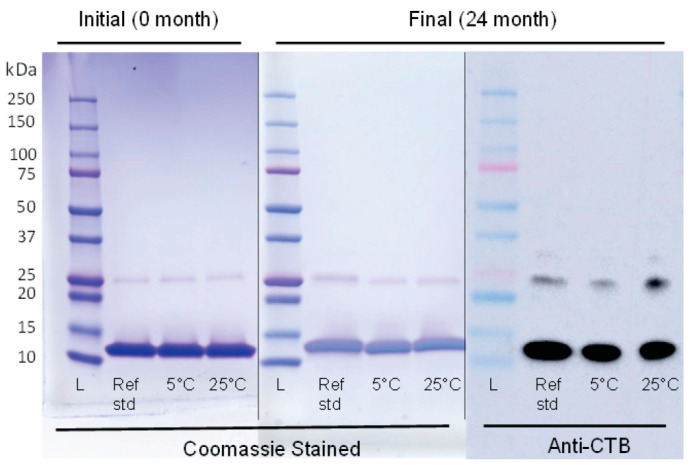
Purity assessment of EPICERTIN DS at 24 months by SDS-PAGE and WB. Coomassie-stained SDS-PAGE is shown for initial (0 month) and final (24 month) time points. An anti-CTB WB of SDS-PAGE shows EPICERTIN DS at the final timepoint probed with 9F9C7 mAb [24]. The expected size of denatured EPICERTIN monomer is ~12.3 kDa. Additional bands present are seen at ~25 kDa, indicating the presence of EPICERTIN dimer.

**Figure 3 pharmaceutics-17-00259-f003:**
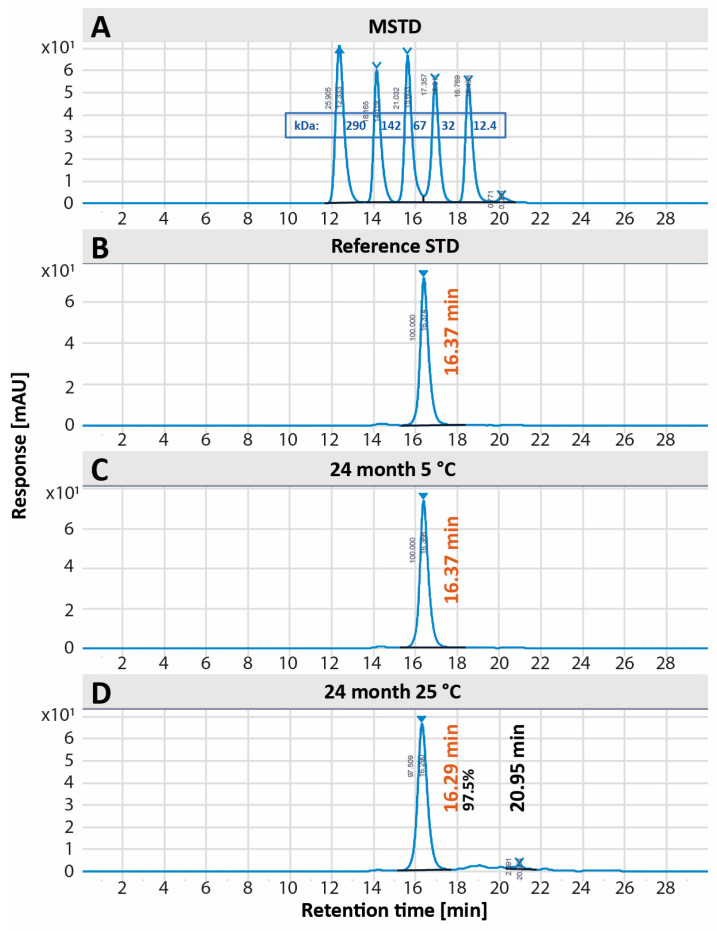
SEC-HPLC purity assessment of EPICERTIN DS at 24 months. The SEC-HPLC chromatograms demonstrate the identity of EPICERTIN pentamer (61.4 kDa). (**A**) Molecular weight standard. The retention times of the peaks from the reference standard (**B**) and samples (**C**,**D**) correspond with the correct size range as compared to the molecular weight standard (**A**). The assay acceptance criteria (≥95% pentamer) were met at both (**C**) 5 °C (100% pentamer) and (**D**) 25 °C/60% RH (97.5% pentamer) at the final time point.

**Figure 4 pharmaceutics-17-00259-f004:**
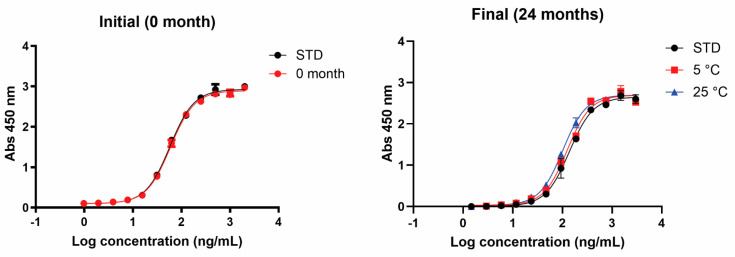
Potency assessment of EPICERTIN DS at 24 months by GM1/KDEL ELISA. (**Left**) Initial time point potency assessment. (**Right**) At 24 months, compared with the reference standard (STD), both 5 °C and 25 °C/60% RH are within specification (<±30% EC_50_ shift).

**Figure 5 pharmaceutics-17-00259-f005:**
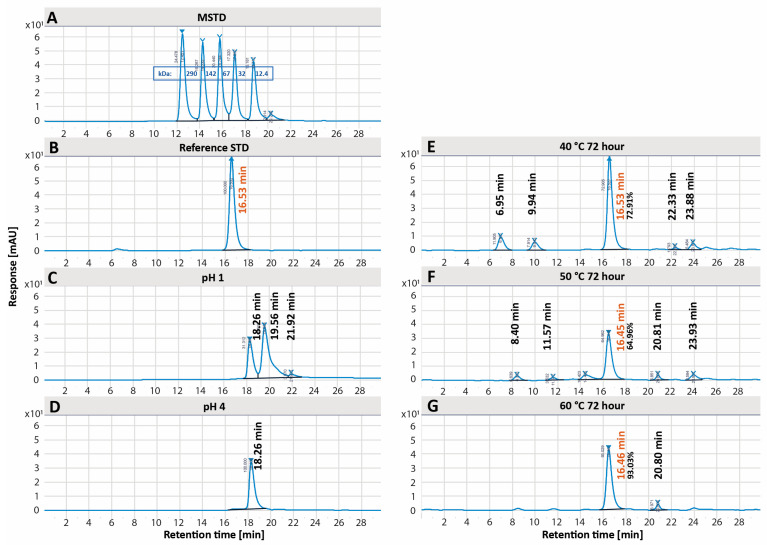
Forced degradation conditions that failed to meet specifications by SEC-HPLC. (**A**) Molecular weight standard. (**B**) Reference standard. (**C**,**D**) Low pH: pH 1 and pH 4, respectively. (**E**–**G**) Representative chromatograms of high temperature conditions: 40 °C 72 h, 50 °C 72 h, 60 °C 72 h, respectively.

**Figure 6 pharmaceutics-17-00259-f006:**
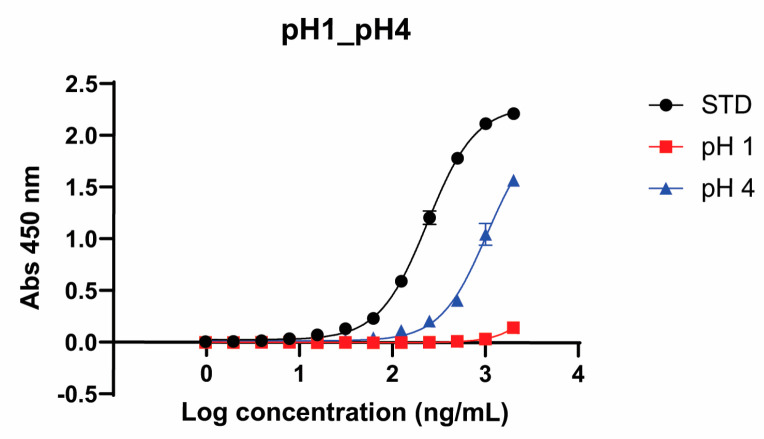
Forced degradation conditions that failed to meet specifications by GM1/KDEL ELISA potency assessment. GM1 binding is attenuated with pH 4 (335.92% EC_50_ shift) and negligible with pH 1 (4128.50% EC_50_ shift).

**Figure 7 pharmaceutics-17-00259-f007:**
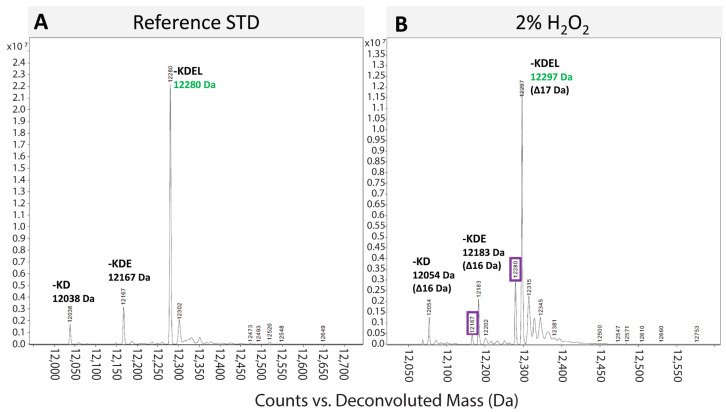
ESI-MS scan of oxidized EPICERTIN DS. (**A**) Reference standard chromatogram demonstrating EPICERTIN DS identity within specification (122,79.9 ± 3 Da) as well as expected -KD and -KDE truncated species. (**B**) Oxidized EPICERTIN DS under 36 h 2% H_2_O_2_ treatment showing expected species albeit with 16 and 17 Da mass increases which indicate singlet oxidation. Residual unoxidized species are outlined in purple.

**Figure 8 pharmaceutics-17-00259-f008:**
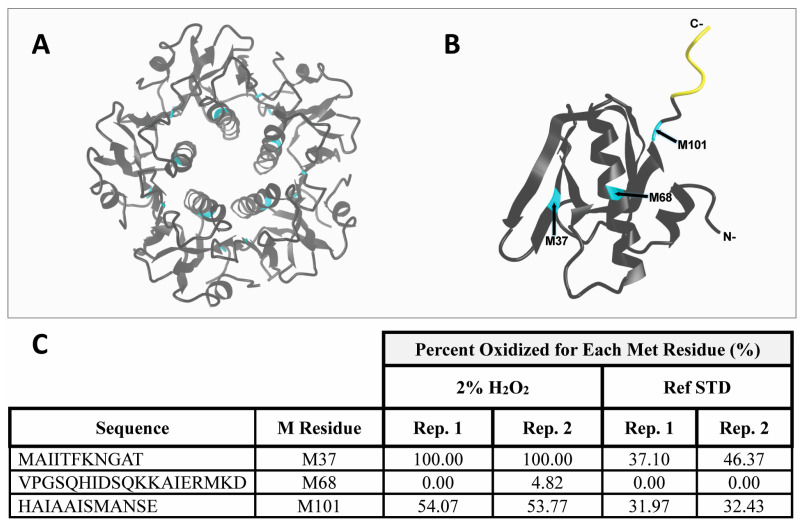
Determination of the oxidation site of EPICERTIN DS upon 36 h exposure to 2% H_2_O_2_. (**A**) Theoretical rendering of EPICERTIN pentamer highlighting the three Met residues (cyan) per monomer. (**B**) Theoretical rendering of EPICERTIN monomer highlighting Met residues (cyan) and EPICERTIN’s C-terminal -SEKDEL extension (yellow). PDB files were generated using AlphaFold v2.3.2 [28], with annotations made using iCn3D. (**C**) Peptide mapping revealed Met37 is 100% oxidized and Met101 is 54% oxidized upon 36 h exposure to 2% H_2_O_2_.

## Data Availability

The data presented in this study are available within the article and its Appendix A.

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
