# Peer review of "Preclinical Long-Term Stability and Forced Degradation Assessment of EPICERTIN, a Mucosal Healing Biotherapeutic for Inflammatory Bowel Disease"

_pharmaceutics, 2025, doi:10.3390/pharmaceutics17020259_

Round 1
Reviewer 1 Report (Previous Reviewer 2)
Comments and Suggestions for Authors
Reviewer # 1
This manuscript is a revised version.
The authors have provided an appropriate explanation about the different questions, and I am in agreements with the author’s reply.
In addition, the quality and interest of the paper have improved with the rest of suggestion given by other reviewers. In this Reviewer’s opinion, the manuscript is suitable for publication in the Pharmaceutics Journal in present form.
Reviewer 2 Report (Previous Reviewer 3)
Comments and Suggestions for Authors
I would like to thank the authors for taking all my comments and suggestions into consideration. I believe that the manuscript has good merit and includes information and data that can be helpful to the general filed of this study. Also, the experiments are well designed and the data are adequately presented and supported.
This manuscript is a resubmission of an earlier submission. The following is a list of the peer review reports and author responses from that submission.
Round 1
Reviewer 1 Report
Comments and Suggestions for Authors
The manuscript supplied for review "Long-Term and Forced Degradation Stability Assessments of Epicertin, a Mucosal Healing Biotherapeutic for Inflammatory Bowel Disease" details the long-term and forced degradation stability assessments of EPICERTIN, a mucosal healing biotherapeutic for inflammatory bowel disease (IBD). The results indicate that Epicertin remains stable over two years when stored in phosphate-buffered saline (PBS) at 5°C, and the forced degradation study successfully identified its degradation products, highlighting its stability and potential as a therapeutic candidate.
The authors should be commended for a novel exploration of the subject, in particular the comprehensive level of analysis performed. However, there are a few areas the author may wish to address prior to publication.
Introduction:
While the manuscript discusses the potential of Epicertin in inflammatory bowel disease (IBD), it would benefit from a more thorough explanation of the current therapeutic landscape for mucosal healing agents. For example, expanding on disadvantages of current treatments and and how Epicertin's mechanism of action (e.g., binding to GM1 ganglioside, activating KDEL receptor) differentiates it from existing therapies could strengthen the case for its novelty.
The manuscript could also benefit from a more in depth discussion on the regulatory requirements for stability testing. As currently presented in the manuscript it does now indicate that the authors have a full understanding of the process ..."a battery of testing" is a particularly clumsy description.
Finally within the introduction the authors could highlight the benefits of a successful treatment i.e reducing surgery and further complications arising from prolonged IBD exposure such as chances of developing colorectal cancer.
Materials and Methods:
My personal preference is for a more structured approach to this section i.e appropriate section numbering.
While the manuscript covers various forced degradation conditions, it lacks details on how these conditions were selected. A deeper discussion of the physiological relevance (e.g., how temperature stress mimics real-world storage or transport conditions) would enhance the reader’s understanding of the practical implications.
Discussion:
In the discussion of the protein concentration deviations due to container closure system issues at 25°C/60% RH (Line 423 onwards), propose potential solutions, such as more robust packaging methods. Discussing how these challenges will be addressed in future clinical development or large-scale production could demonstrate preparedness for moving toward clinical trials. Could this behaviour have been anticipated before the study and accounted for during the study.
Comments on the Quality of English LanguageOne of my first impressions from the entire manuscript is why Epicertin is written throughout in uppercase letters.
Some sections switch between past and present tense unnecessarily, e.g., "EPICERTIN remains stable for 2 years" (present tense) vs. "was subjected to a study" (past tense). Keeping tense consistent improves the clarity of the whole manuscript.
In the sentence "The treatment regimen for UC and CD are similar," "regimen" is singular, so it should be "is similar." or regimens ...are similar
Care should be taken regarding font choice as there appears to multiple font sizes in use throughout the manuscript.
Reviewer 2 Report
Comments and Suggestions for Authors
This paper describes and illustrate the long-term and forced degradation stability assessments of Epicertin, a mucosal healing biotherapeutic for inflammatory bowel disease. The purpose of the study is well explained in the introduction and appears to be adequately referenced.
However, in opinion of this reviewer there is a discrepancy in the results (data interpretation) and the conclusions. (1) This discrepancy is related with the design of the experiments, especially under forced degradation conditions and (2) The problems detected at the 9-month timepoints where the authors detected a problem with the samples stored at 25ºC due to an inadequate container-closure system. In this context, all the results and conclusions are no valid.
Appropriate statistical analysis should be applied, when necessary, to quantitative data reported. The methods of analysis, including justification and rationale, should be described fully. These descriptions should be sufficiently clear to permit independent calculation of the results presented. In this point, any information about this was provided.
With respect the acceptance criteria established for drug concentration 1.0±0.2 mg; this needs an explanation (lines 201-203). This value supposes a coefficient of variation of 20%, whereas the specification limits to establish the release limits and shelf- life are 95% and 105% of the nominal concentration (see ICHQ1A(R2) and ICHQ1E). In addition, what is the CV of the analytical method used to determine the drug concentration? Any data about the validation was reported (Why?)
With respect to the point #1. The experiments at 5º and 25ºC were carried out by using a formulation of 1.06 mg/mL in a vial. This is ok. However, the experiments under accelerated conditions used an aliquot of this concentration stored at 40ºC in a 2 ml cap vial (for 40ºC, the authors used 20µ of Epicertin DS 1.06 mg/mL at 2ml cap vial). This is not correct since this implies a change of the conditions, (different concentrations) and the degradation kinetic could depends on the concentration. Any information on degradation and kinetics of degradation was provided. The authors only reported the identity of the degradation products detected, but they must verify this behaviour also will be observed under real conditions.
In opinion of this reviewer, the manuscript is not suitable for publication in the Pharmaceutics Journal.
Reviewer 3 Report
Comments and Suggestions for Authors
This work investigates the long-term stability and degradation of EPICERTIN under certain stress environments. The goal is to evaluate whether it could be potentially used as a biotherapeutic for mucosal healing in inflammatory bowel disease and other mucosal disorders. The authors have done a great work with the experimental design, the experiments themselves have a systematic approach, and the study overall is very cohesive. Also, the clarity of presentation is very good, and the conclusions are well supported by the data generated. I only have some minor comments and recommendations for the authors.
Title
Just a recommendation for the authors regarding the title. I would rephrase the first part of the title as follows “Long-Term Stability and Forced Degradation Assessment of EPICERTIN, …”. I come from the world of accelerating aging and long-term stability of polymers and biomaterials, and usually we refer to the degradation of materials as “long-term stability”, although we study their degradation. Also, the words degradation and stability right next to each other in the title does not read well. Again, this is just a recommendation.
Abstract
Line 18: Please add space between the number and the units i.e., 25 °C.
Keywords
Line 33: I would replace the work “stability assessment” with “long-term stability”. I believe is a more appropriate term for a 2-year study.
Introduction
Line 43: The very last part of this sentence is not necessary; authors could re-write as follows “…when agents loose or lack efficacy [2-4].”.
Lines 71- 76: These last two sentences need citations to support the claims.
Line 78: I am a big proponent of the Oxford comma, so I would add one after the word “manufacturing” and before the “and” in this sentence. Just a recommendation.
Line 96: Please re-write the sentence as follows “…be tested for stability at 25 °C and 60% relative humidity (RH).”. The term “at minimum” is not required here, as you are proposing additional studies in different storage environments in the next sentence.
Line 102: Please combine the two sentences as follows “…the understanding of its molecular properties [17,18], and provide crucial information necessary for…”.
Materials & Methods
When concentration of solutions is not expressed in mg/mL, normality, or molarity, but % concentration is used, please specify if it is by volume or by mass. Alao, please consider moving the sub-section that discusses production of EPICERTIN in the beginning of the Materials & Discussion section.
Line 125: Please add space between the number and the units i.e., 200 V.
Line 131: Please fix the inequality symbol the formatting is not correct.
Line 167: Please add space between the number and units i.e., 23 °C. I would advice the authors to review the entire manuscript and add space between numbers and units, as need.
Results
Table 1: How do we know that the value of 1.35 mg/mL at 24 months (25 °C/60% RH) is “Fail”? Was there a student t-test performed to evaluate whether the value is within statistical error? The values for concentration and pH need standard error.
Figure 1: The values in the x- and y-axis are very small and hard to read. I need to zoom in to 300% to barely see them and at this point the resolution is poor. I would strongly advice the authors to replot these graphs with larger fonts for the values. Also, the parameters used for data collection are not considered necessary information, so please remove from the graphs. Finally, be consistent with the use of Dalton (Da) and atomic mass unit (amu) in the figure. Your peak labels within the graphs have Da and so does the figure caption, whereas the x-axis is labeled as “Deconvoluted Mass (amu)”. I know it is the same unit, but try to be consistent, at least within the same figure. Same for figure 7.
Figures 3 & 5: The values in the x- and y-axis are very small and hard to read, please consider using larger fonts.
Line 384: This is the first time you talk about cysteine (Cys) and methionine (Met) in the manuscript, please define the abbreviations here.
Discussion
Line 423: The authors discuss the potential evaporation of PBS during the storage conditions from the storage vails as a plausible explanation to the observed increase in concentration after 24 months (Table 1). Was dehydration/lyophilization of EPICERTIN investigated as a potential long-term storage solution, along with some rehydration and evaluation studies? This is just a question. If you have done such studies, it would be nice to add some of the findings here for comparison. If not, maybe something to investigate in the future.